# Weightbearing Imaging Assessment of Midfoot Instability in Patients with Confirmed Hallux Valgus Deformity: A Systematic Review of the Literature

**DOI:** 10.3390/diagnostics14020193

**Published:** 2024-01-16

**Authors:** Grayson M. Talaski, Anthony N. Baumann, Bshara Sleem, Albert T. Anastasio, Kempland C. Walley, Conor N. O’Neill, Samuel B. Adams

**Affiliations:** 1Department of Orthopedics and Rehabilitation, University of Iowa, Iowa City, IA 52242, USA; 2College of Medicine, Northeast Ohio Medical University, Rootstown, OH 44272, USA; abaumann@neomed.edu; 3College of Medicine, American University of Beirut, Beirut 1107-2020, Lebanon; bms26@mail.aub.edu; 4Department of Orthopaedic Surgery, Duke University, Durham, NC 27708, USA; albert.anastasio@duke.edu (A.T.A.); conor.n.oneill@duke.edu (C.N.O.); samuel.adams@duke.edu (S.B.A.); 5Department of Orthopaedic Surgery, University of Michigan, Ann Arbor, MI 48109, USA; kcwalley@med.umich.edu

**Keywords:** hallux valgus deformity, midfoot instability, WBCT, systematic review, imaging

## Abstract

Hallux valgus deformity (HVD) involves subluxation of the first metatarsophalangeal joint. While HVD is primarily considered a forefoot condition, midfoot instability may play a significant role in its development and severity. However, very few studies have placed a heavy emphasis on studying this phenomenon. Therefore, this review had a particular focus on understanding midfoot instability based on weightbearing imaging assessments of the TMT joint. This review followed Preferred Reporting Items for Systematic Reviews and Meta-Analyses (PRISMA) guidelines and searched five databases for studies on midfoot instability in HVD patients. The severity of HVD was defined by hallux valgus angle (HVA) and distal metatarsal articular angle (DMAA). Data was extracted, and articles were graded using the Methodological Index for Non-Randomized Studies (MINORS). Of 547 initially retrieved articles, 23 met the inclusion criteria. Patients with HVD showed higher HVA and DMAA on weightbearing radiographs (WBRG) and weightbearing computed tomography (WBCT) compared to healthy individuals. Midfoot instability was assessed through intermetatarsal angle (IMA) and tarsometatarsal angle (TMT angle). Patients with HVD exhibited greater IMA and TMT angles on both WBRG and WBCT. This review highlights the importance of weightbearing imaging assessments for midfoot instability in HVD. IMA and TMT angles can differentiate between healthy individuals and HVD patients, emphasizing the significance of midfoot assessment in understanding HVD pathology. These findings validate the limited evidence thus far in the literature pertaining to consistent midfoot instability in HVD patients and are able to provide ample reasoning for physicians to place a larger emphasis on midfoot imaging when assessing HVD in its entirety.

## 1. Introduction

Hallux valgus deformity (HVD), a common foot and ankle condition [1,2], relates to a medial shift of the first metatarsal head followed by a lateral shift of the proximal phalanx [3]. As HVD progresses, subluxation of the first metatarsophalangeal (MTP) joint is observed [1]. HVD is predominantly described as a forefoot condition, but biomechanical failure or hereditary conditions at any point along the first ray can lead to hallux valgus [3]. Therefore, measurements that assess the entire first ray, such as intermetatarsal angle [4,5,6,7,8,9], may describe the condition more comprehensively. While the intermetatarsal angle is often measured for patients with HVD, evidence is lacking to conclude that physicians factor its value into decisions. Studies have eluded that hypermobility of the first tarsometatarsal (TMT) is related to HVD severity [4,10,11,12,13], but the number of studies analyzing solely midfoot instability in HVD patients is slim. However, midfoot assessments of HVD have allowed for a more comprehensive assessment of HVD, as various TMT measurements may directly lead to changes in corresponding MTP measurements [14]. For example, the TMT angle taken from a sagittal view has shown significant correlations to HVD progression, likely due to increased dorsiflexion at the MTP joint [14]. Furthermore, the reliability of forefoot measurements associated with HVD, such as hallux valgus angle, has shown greater variability than midfoot-related measurements [15]. Therefore, it is of great interest to study HVD from the midfoot, as the entirety of the first ray mobility can be more completely assessed. 

While there are previous systematic reviews pertaining to HVD, topics are limited to demographic [16], treatment [17], and outcome-based studies [18]. Furthermore, no review dedicated solely to analyzing HVD imaging-based studies has been performed, particularly in weightbearing. As a proper imaging assessment may lead to a more complete pre-operative assessment, a review focused on radiographic assessment of HVD is of high interest. Therefore, the primary aim of this systematic review was to summarize all studies pertaining to HVD, with a particular focus on understanding midfoot instability based on weightbearing imaging assessments of the TMT joint.

## 2. Materials and Methods

### 2.1. Study Creation

This study is a systematic review examining midfoot instability in patients with confirmed HVD assessed via weightbearing imaging modalities to further explore the understanding of how HVD impacts the midfoot, potentially guiding treatment in the future. This systematic review was performed in line with the most recent Preferred Reporting Items for Systematic Reviews and Meta-Analyses (PRISMA) guidelines for optimal clarity [19]. Five databases—PubMed, SPORTDiscus, CINAHL, MEDLINE, and Web of Science—were searched from database creation until 6 July 2023. The search algorithm used in each of the five databases to address our study objective was (hallux valgus OR bunion) AND (medial column OR midfoot OR tarsometatarsal OR “tarsal-metatarsal” OR “metatarsal-cuneiform” OR Lisfranc) AND (mobility OR instability OR stability OR rotation). 

### 2.2. Inclusion and Exclusion Criteria

Inclusion criteria were articles that examined the midfoot region, articles that had patients with a diagnosis of HVD, articles that were full-text, articles that were in English, and randomized controlled trials or observational studies with at least ten patients. Ten patients were chosen as a patient threshold to prevent low-power studies from potentially diluting the findings. Exclusion criteria were articles not examining the TMT joint/midfoot, articles that only had healthy patients or patients without a diagnosis of HVD, articles not having full-text or an abstract, articles not in English, systematic reviews, meta-analyses, books, case reports, case series with nine patients or less, and clinical commentaries. Articles that involved surgical correction to HV were included if they reported preoperative (i.e., non-surgical) values that could be used to assess the impact of the HV on the midfoot region. As this study was solely focused on HVD midfoot instability, post-operative measurements were not relevant.

### 2.3. Study Definitions

As several studies reported both healthy patients as well as patients with HVD, it was necessary to use measurements to define the severity of HVD in this study. The severity of HVD was defined in this study via hallux valgus angle (HVA) and distal metatarsal articular angle (DMAA). Furthermore, the main objective of this study was to examine the severity of midfoot instability and pathology. For the purposes of this study, the severity of midfoot instability was primarily defined via measurements such as intermetatarsal angle (IMT angle) and tarsometatarsal angle (TMT angle) in various imaging views. Weightbearing imaging was defined as weightbearing radiograph (WBRG) or computed tomography (WBCT). 

### 2.4. Article Screening Process

After all five databases were searched with the aforementioned algorithm, all of the retrieved articles were downloaded into Rayyan, an online public software commonly utilized in the literature to allow for efficient article screening [20]. Duplicate articles were first removed manually, and then all remaining articles were screened by title and abstract via the inclusion and exclusion criteria. Article screening was performed by multiple authors. After screening by title and abstract, articles were screened by full-text for final article inclusion. 

### 2.5. Data Extraction

Data extraction was performed by a single author. Data extracted from the included articles include first author, year of publication, type of study, number of patients, number of feet, average patient age, type of imaging modality (weightbearing radiograph or weightbearing CT), HVA (degrees), IMT angle (degrees), TMT angle (degrees), sagittal lift (mm), and Meary’s angle (degrees). 

### 2.6. Article Quality Grading

All observational studies were graded via the Methodological Index for Non-Randomized Studies (MINORS) as previously used in the literature for systematic reviews and meta-analyses [21]. The MINORS scale differentiates between comparative and non-comparative studies, with comparative studies being out of 0–24 points and non-comparative studies being out of 16 points. Each item on the MINORS scale assesses the quality of the article and is worth 0–2 points. All grading was completed by one author. 

### 2.7. Statistical Analysis 

The Statistical Package for the Social Sciences (SPSS) version 29.0 (Armonk, NY, USA: IBM Corp) was used for analysis in this systematic review. Descriptive statistics (means, frequencies) and frequency-weighted means were utilized to report the data. Due to the heterogeneity of the data, a narrative approach to systematic review with qualitative statistics was used as meta-analysis could not be performed. 

## 3. Results

### 3.1. Initial Study Results

A total of 23 articles met the inclusion criteria from 547 articles initially retrieved from the five databases utilized in this systematic review [14,22,23,24,25,26,27,28,29,30,31,32,33,34,35,36,37,38,39,40,41,42]. Refer to Figure 1 for the PRISMA diagram outlining the search process for this systematic review. 

### 3.2. Article Quality Results

All 23 included articles were graded via the MINORS scale due to the observational nature of the included studies. The mean MINORS score for all included articles (*n* = 23 articles) was 10.4 ± 4.0 points (range: 5.0–20.0 points). Based on study type, the mean MINORS score was 8.4 ± 1.9 points (range: 5.0–12.0 points) for non-comparative studies and 16.2 ± 2.0 points (range: 14.0–20.0 points) for comparative studies. Refer to Table 1 for more specific information on the MINORS grading for each individual article included in this study.

### 3.3. General Patient Demographics

Total patients (*n* = 962) had a frequency-weighted mean age of 45.5 ± 6.4 years (*n* = 962; 100% of patients reported). However, three studies did not report the number of patients but reported the number of feet investigated in their study. Furthermore, 20 out of 23 articles reported on the number of feet investigated, with a total of 1232 feet included in this systematic review. Based on the patient subgroup, there were 851 patients with HV and 111 healthy patients without HV used as comparison groups in some of the included studies. The frequency-weighted mean age of patients with HV (*n* = 851) was 45.6 ± 6.4 years, and the frequency-weighted mean age of healthy patients without HV (*n* = 111) was 44.2 ± 6.3 years. Of the 1232 feet included in this study, 1057 feet belonged to patients with HV, and 175 feet belonged to healthy patients without HV. In terms of imaging modality, 196 patients (20.4%) were evaluated via WBCT, and 766 patients (79.6%) were evaluated using WBRG. Refer to Table 2 for more specific information on the demographics and patient information for each individual included in the article.

### 3.4. Severity of Hallux Valgus by Imaging 

For the severity of HV as defined by HVA and/or DMAA, patients with HV evaluated with WBCT (*n* = 185 feet) had a frequency-weighted mean HVA (axial view) of 31.2 ± 1.4 degrees (*n* = 112; 60.5% of feet reported), a mean HVA (coronal view) of 28.6 degrees (*n* = 10 feet; 5.4% of feet reported) and a frequency-weighted mean HVA (sagittal view) of 33.1 ± 5.8 degrees (*n* = 41; 22.2% of feet reported). Healthy patients without HV evaluated with WBCT (*n* = 175 feet) had a frequency-weighted mean HVA (axial view) of 10.5 ± 1.8 degrees (*n* = 109; 62.2% of feet reported), a mean HVA (coronal view) of 11.0 degrees (*n* = 36 feet; 20.6% of feet reported) or a mean HVA (sagittal view) of 14.1 degrees (*n* = 10; 5.7% of feet reported). For severity of HV as defined by HVA and/or DMAA, patients with HV evaluated by WBRG (*n* = 872 feet) had a frequency-weighted mean HVA (anterior-posterior) of 28.9 ± 6.7 degrees (*n* = 673; 77.2% of feet reported). The frequency-weighted mean DMAA on WBRG (*n* = 263 feet) was 14.7 ± 4.5 degrees for patients with HV. Refer to Table 2 for more specific information on the demographics and patient information for each individual included in the article. 

### 3.5. Midfoot Instability via Intermetatarsal Angle 

Kimura et al. (2017) reported a significantly greater TMT angle (sagittal) on WBCT in patients with HV as compared to healthy patients without HV (22.1 versus 9.3 degrees; *p* < 0.01) [36]. Similarly, Randich et al. (2021) reported a significantly greater TMT angle (frontal) on WBCT in patients with HV as compared to healthy patients without HV (16.5 versus 8.7 degrees; *p* < 0.001) [40]. Both Lee et al. (2022) and Ji et al. (2023) reported significantly greater IMT angles (axial view) on WBCT in patients with HV as compared to healthy patients without HV (*p* < 0.001 and *p* < 0.01, respectively) [14,27]. The frequency-weighted mean TMT angle (axial) on WBCT was 15.4 ± 1.0 degrees for patients with HV (*n* = 112 feet) and 8.6 ± 0.5 degrees for healthy patients without HV (*n* = 109 feet). The frequency-weighted mean IMT angle (anterior-posterior) on WBRG for patients with HV (*n* = 763 feet) was 15.2 ± 2.7 degrees. Refer to Table 3 for more information on IMT angles from individual articles included in this systematic review.

### 3.6. Midfoot Instability via Tarsometatarsal Angle

The frequency-weighted mean TMT angle (sagittal view) on WBCT was 1.9 ± 1.4 degrees for patients with HV (*n* = 122 feet) as compared to 0.9 ± 0.7 degrees in healthy patients without HV (*n* = 119 feet). From individual articles, both Kimura et al. (2017) and Lee et al. (2022) found significantly larger TMT angles (sagittal view) on WBCT in patients with HV compared to healthy patients without HV (*p* < 0.01 and *p* < 0.001, respectively) [14,36]. Likewise, Ji et al. (2023) reported a significantly larger TMT angle (sagittal view) in patients with HV as compared to healthy patients without HV on WBCT (1.6 versus 0.9 degrees; *p* < 0.01) [27]. From a different view, Randich et al. (2021) reported a TMT angle (frontal view) on WBCT of −5.36 ± 6.28 degrees in patients with HV as compared to −1.28 ± 6.33 degrees in healthy patients without HV (*p* = 0.08) [40]. On WBRG, King et al. (2004) reported higher absolute TMT angles (anterior-posterior and lateral view) of 11.0 ± 7.0 degrees and 13.0 ± 8.0 degrees in patients with HV as compared to TMT angles (anterior-posterior and lateral view) of 8.0 ± 4.0 degrees and 4.0 ± 8.0 degrees in healthy patients without HV [29]. Refer to Table 3 for more information on TMT angles from individual articles included in this study.

## 4. Discussion

In this systematic review, an analysis of the midfoot region of the foot was performed for patients with confirmed HVD. To the authors’ knowledge, this is the first review to study HVD from an image-based perspective, as all previous reviews have been centered around prevalence and outcome-based topics. While these topics are of high interest, a review focused strictly on pre-operative imaging is of great interest, as proper pre-operative radiographic assessment is essential for surgical planning [44]. While a review focused on the MTP joint would be insightful, a midfoot assessment was chosen for this review for two reasons. (1) Increased TMT instability and first ray hypermobility has been linked to increased HVD severity [14,25,45], and (2) the recent emergence of weightbearing computed topography (WBCT) allows for three-dimensional assessment of the entire first ray [46,47,48], allowing for measurements that are difficult on pain radiographs. Furthermore, MTP joint measurements of HVD indicate the severity of the resulting deformity, not necessarily revealing the cause of the deformity. As instability at any point along the first ray can cause HVD [3], understanding the impact that the midfoot has on the pathology of HVD was of high interest for this review. As no direct comparison of measurements across studies was possible, this review’s purpose remains to summarize the importance of weightbearing and midfoot assessment when planning for HVD correction.

The most common midfoot measurements to analyze HVD were intermetatarsal angle (IMA) and tarsometatarsal angle (TMT angle). Each included study was able to differentiate healthy and HVD using these measurements, regardless of image modality. As WBCT has yet to become universal care for foot and ankle clinics, this conclusion was encouraging for WBRG. However, while WBRG manual measurements have demonstrated reliability [49], two-dimensional analysis of HVD may lead to less detailed pre-operative planning due to HVD often including a rotational aspect of the first ray [50]. As improper correction of first ray/medial column rotation is tied to poor HVD post-operative outcomes, accurate measurement of midfoot parameters is of high priority [51]. Fortunately, studies that compare the rotational measurement sensitivity of WBRG and WBCT have found agreement between the two image modalities [52,53]. However, WBCT has shown increased sensitivity when assessing rotation, likely due to its inherent three-dimensional advantage [53]. One concern still surrounds the reliability and consistency of the WBRG scan protocol when assessing these measurements [15]. While no study has found significant differences for TMT angles in lateral views, the dorsoplantar view has been shown to differ significantly with varying scan protocols [15]. It is interesting to note that differences in scan protocol led to greater variability for traditional WBRG forefoot measurements than WBRG midfoot measurements, suggesting that midfoot analysis is not only potentially more comprehensive but also more reliable regardless of scan protocol [15]. While some may suggest that WBCT scan protocol may also vary significantly due to WB being painful for severe deformities, there is little to no evidence suggesting that HVD causes significant pain in a static stance.

While this review described the severity of injury based upon a forefoot measurement (HV angle), evidence exists suggesting that forefoot measurements are a direct result of medial column/first ray alignment [14]. Even though HVD is most evident at the metatarsal head [3], data suggests that patient-specific anatomy and biomechanics of the TMT joint may have an impact on the emergence of HVD [14]. This is due to the TMT joint being the apex of the metatarsal, as well as being the center of rotation of angulation (CORA) [54]. Numerous studies point to this joint being the true center of HVD and indicate that correction of the TMT joint addresses the primary deformity while also preventing the development of secondary deformities that may form in part due to MTP arthrodesis to correct HVD [55,56,57,58,59,60]. While this review was unable to statistically assess the sensitivity of TMT measurements directly, one takeaway revolved around the lack of standard TMT measurements. Two measurements to assess an entire three-dimensional deformity are unlikely to provide a complete pre-operative assessment. Furthermore, the method of measurement collection was primarily manual measurement. With the recent advancements in semi-automatic and automatic measurement capabilities, the possibility exists to not only limit variability but also provide more standardized, comprehensive pre-operative assessment [46]. Future research and standardization of TMT-related measurements to assess HVD may lead to a more detailed pre-operative plan.

One crucial aspect of this review was its focus on weightbearing. In the included studies that directly compared NWB and WB, significant differences were found between patients with HVD and healthy controls [35]. While it may seem obvious to assess HVD with WB imaging, comprehensive WB assessment is not standard. Many physicians rely on intra-operative assessment of first ray rotation when correcting HVD [52]. As only simulated WB can be obtained within the operating room, and WB changes first ray metrics [61], there should be an emphasis on consistent, standard WB pre-operative assessment. 

Regarding the limitations of this study, the lack of direct comparison between studies is of primary concern. Only two common midfoot measurements were collected across all studies, suggesting that future research should work towards additional standard midfoot measurements. This improvement could allow for meta-analysis of midfoot measurements, providing significant guidance for physicians when deciding which metrics to rely on for pre-operative planning. While this was mentioned previously, the lack of scan protocol standardization across studies is nearly impossible to account for. As changes in WB cause changes in midfoot measurements [61], it is important that future research also works to account for differences in scan acquisition. As pain may prevent patients from standing with an exact 50/50 load distribution, comparison across studies requires a consistent scan protocol. Furthermore, the studies included in this systematic review were of an observational nature, indicating that bias likely impacted the results of this manuscript. Future research should focus on higher level-of-evidence studies to further solidify the impact of WB imaging on the assessment of midfoot instability in patients with HVD. 

## 5. Conclusions

In conclusion, this systematic review focused on WB imaging assessments of midfoot instability in patients with HVD. To the authors’ knowledge, this is the first comprehensive review of HVD from a pre-operative, image-based perspective, emphasizing the importance of midfoot assessment in understanding the pathology of HVD. The review highlighted the significance of measurements such as intermetatarsal angle (IMA) and tarsometatarsal angle (TMT angle) in differentiating between healthy individuals and those with HVD, regardless of the imaging modality used. While weightbearing radiographs (WBRG) have shown promise in assessing midfoot instability, weightbearing computed tomography (WBCT) offers a three-dimensional advantage, particularly when evaluating rotational aspects of the first ray. The review also underscored the potential for more standardized and comprehensive pre-operative assessments with the development of semi-automatic and automatic measurement capabilities for TMT-related measurements. Furthermore, the importance of consistent weightbearing assessment in pre-operative planning was emphasized. This review provides valuable evidence that verifies the little literature surrounding midfoot instability in conjunction with HVD. Future clinical care should place a large emphasis when not only on the diagnosis of HVD but also on assessing the severity of the deformity.

## Figures and Tables

**Figure 1 diagnostics-14-00193-f001:**
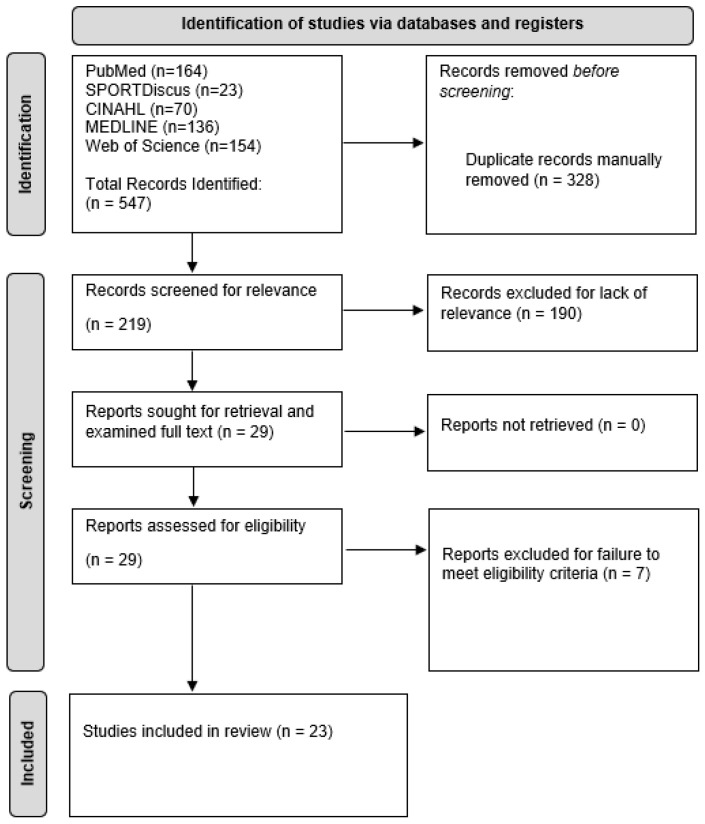
The Preferred Reporting Items for Systematic Reviews and Meta-Analyses (PRISMA) diagram for this systematic review depicting the search process.

**Table 1 diagnostics-14-00193-t001:** The Methodological Index for Non-Randomized Studies (MINORS) grading results for the included articles in this systematic review. Each item is worth 0–2 points for a total of 16 points for non-comparative studies and 24 points for comparative studies.

Author (Year)	Study Type	Total MINORS Score	Clearly Stated Aim	Inclusion of Consecutive Patients	Prospective Collection of Data	End Points Appropriate to Study Aim	Unbiased Assessment of Study End Point	Follow-Up Period Appropriate to Study Aim	Less Than 5% Lost to Follow Up	Prospective Calculation of the Study Size	Adequate Control Group	Contemporary Groups	Baseline Equivalence of Groups	Adequate Statistical Analysis
Conti (2020) [23]	Non-comparative	10	2	2	0	2	1	1	2	0	-	-	-	-
King (2004) [29]	Comparative	15	2	2	1	1	0	0	2	0	2	2	1	2
Ferreyra (2022) [33]	Non-comparative	8	2	2	0	2	0	0	2	0	-	-	-	-
Kernozek (2002) [28]	Non-comparative	6	1	1	0	1	0	1	2	0	-	-	-	-
Dayton (2020) [37]	Non-comparative	9	2	2	0	2	0	1	2	0	-	-	-	-
Lalevée (2022) [31]	Comparative	16	2	1	1	2	0	0	2	0	2	2	2	2
Manceron (2022) [35]	Non-comparative	8	2	2	0	2	0	0	2	0	-	-	-	-
Conti (2022) [22]	Non-comparative	12	2	2	1	2	0	1	2	2	-	-	-	-
Kimura (2017) [36]	Comparative	20	2	2	0	2	2	0	2	2	2	2	2	2
Naguib (2018) [38]	Non-comparative	7	2	1	0	2	0	0	2	0	-	-	-	-
Klemola (2017) [42]	Non-comparative	7	2	1	0	1	0	2	1	0	-	-	-	-
Randich (2021) [40]	Comparative	16	2	2	0	2	0	0	2	2	1	2	1	2
Coughlin (2007) [24]	Non-comparative	10	2	2	0	2	0	2	2	0	-	-	-	-
Thompson (2023) [41]	Non-comparative	9	2	2	0	2	0	1	2	0	-	-	-	-
Ahuero (2019) [32]	Non-comparative	12	2	2	0	2	2	2	2	0	-	-	-	-
Faber (2001) [25]	Non-comparative	6	1	1	0	2	0	0	2	0	-	-	-	-
Lee (2022) [14]	Comparative	14	2	2	0	2	0	0	2	0	1	2	1	2
Oravakangas (2016) [39]	Non-comparative	8	2	2	0	2	0	2	0	0	-	-	-	-
Greeff (2020) [26]	Non-comparative	5	2	1	0	2	0	0	0	0	-	-	-	-
Almaawi (2021) [43]	Non-comparative	9	2	1	0	2	0	2	2	0	-	-	-	-
Kopp (2005) [30]	Non-comparative	8	2	2	0	1	0	1	2	0	-	-	-	-
Ji (2023) [27]	Comparative	16	2	2	0	2	0	0	0	2	2	2	2	2
Ozturk (2020) [34]	Non-comparative	8	2	1	0	2	0	1	2	0	-	-	-	-

**Table 2 diagnostics-14-00193-t002:** Patient demographics for the included articles in this systematic review and meta-analysis. Data recorded included first author, year of publication, type of study, type of patient by group (healthy or patient with hallux valgus (HV)), number of patients, number of feet (due to possible bilateral HV), average patient age (standard deviation and range), imaging modality (weightbearing radiographs (WBRG) or weightbearing computed tomography (WBCT)), and measures of HV severity (hallux valgus angle (HVA) by imaging view and dorsal metatarsal articular angle (DMAA)).

Author (Year)	Study TYPE	Treatment Group	Patients (*n*)	Feet (*n*)	Mean Age (Standard Deviation) (Range)	Imaging Modality	HVA (AP)	HVA (Lateral)	HVA (Axial)	HVA (Sagittal)	HVA (Frontal)	DMAA
Conti (2020) [23]	Retrospective	HV	31	31	51.2 (29–67)	WBCT	-	-	-	29.9 (17–47)	-	-
Lalevée (2022) [31]	Retrospective	Healthy	20	20	37.3 (16.5)	WBCT	-	-	-	-	-	-
		HV	22	22	40.1 (17.4)		-	-	-	-	-	-
Kimura (2017) [36]	Retrospective	Healthy	10	10	56 (5) (50–66)	WBCT	-	-	-	14.1 (2.8)	-	-
		HV	10	10	58 (14.2) (33–74)		-	-	-	43.2 (10.1)	-	-
Randich (2021) [40]	Retrospective	Healthy	36	36	49.31 (12.71)	WBCT	-	-	-	-	11.03 (6.56)	-
		HV	10	10	53.00 (19.35)		-	-	-	-	28.66 (10.99)	-
Lee (2022) [14]	Retrospective	Healthy	30	30	42.97 (17.52)	WBCT	-	-	7.52 (4.49)	-	-	-
		HV	27	30	54.20 (14.01)		-	-	33.50 (9.47)	-	-	-
Ji (2023) [27]	Retrospective	Healthy	-	79	42 (32–51)	WBCT	-	-	11.6 (10.1–14.0)	-	-	-
		HV	-	82	46 (37–55)		-	-	30.4 (22.4–38.6)	-	-	-
Conti (2022) [22]	Retrospective	HV	39	-	51.5 (24.1–64.3)	WBRG	33.2 (10.7)	-	-	-	-	-
						WBCT	-	-	-	-	-	-
King (2004) [29]	Prospective	Healthy	15	-	36 (15) (18–62)	WBRG	5 (3)	-	-	-	-	-
		HV	25	-	48 (17) (14–81)		13 (7)	-	-	-	-	-
Ferreyra (2022) [33]	Retrospective	HV	30	37	45.68 (15–76)	WBRG	32.12	-	-	-	-	-
Kernozek (2002) [28]	Retrospective	HV	25	-	43 (40–60)	WBRG	31.7 (4.7)	-	-	-	-	-
Naguib (2018) [38]	Retrospective	HV	-	59	-	WBRG	11.59 (3.79)	-	-	-	-	-
Klemola (2017) [42]	Retrospective	HV	66	84	47.9 (10.2)	WBRG	30.1 (7.0)	-	-	-	-	-
Coughlin (2007) [24]	Retrospective	HV	103	122	50 (22–78)	WBRG	30 (20–53)	-	-	-	-	10 (0–20)
Thompson (2023) [41]	Retrospective	HV	77	90	48.8 (16.2)	WBRG	-	-	-	-	-	-
Ahuero (2019) [32]	Retrospective	HV	13	14	56 (22–75)	WBRG	32 (26.5–41)	-	-	-	-	-
Faber (2001) [25]	Prospective	HV	94	109	41.4 (15–63)	WBRG	-	-	-	-	-	-
Oravakangas (2016) [39]	Retrospective	HV	20	23	50 (22–69)	WBRG	38 (5)	-	-	-	-	-
Greeff (2020) [26]	Retrospective	HV	23	32	43 (20–68)	WBRG	33 (16–46)	-	-	-	-	16 (4–26)
Almaawi (2021) [43]	Retrospective	HV	89	100	40.7	WBRG	33.2 (8.0)	-	-	-	-	-
Kopp (2005) [30]	Retrospective	HV	29	34	54.2 (27–84)	WBRG	33.6 (17–61)	-	-	-	-	-
Ozturk (2020) [34]	Prospective	HV	10	10	59.3 (15.8) (25–72)	WBRG	38.4 (6.5)	-	-	-	-	-
Manceron (2022) [35]	Retrospective	HV	-	20	-	WBRG	32	-	-	-	-	-
		HV	-	20	-		34.2	-	-	-	-	-
		HV	-	9	-		37.9	-	-	-	-	-
Dayton (2020) [37]	Retrospective	HV	108	109	33.9 (14.1)	WBRG	22.9 (7.6)	-	-	-	-	19.6 (9.2)

**Table 3 diagnostics-14-00193-t003:** Imaging assessments of midfoot instability in patients with HV and healthy patients without HV in the individual articles included in this study. Data recorded includes first author, year of publication, patient group (patients with hallux valgus (HV) or healthy patients), number of patients, number of feet, intermetatarsal angle (IMT angle), tarsometatarsal angle (TMT angle) by view, sagittal lift (in millimeters), and Meary’s angle.

Author (Year)	Treatment Group	Patients	# Feet	IMT Angle (AP)	IMT Angle (Lateral)	IMT Angle (Axial)	IMT Angle (Sagittal)	IMT Angle (Frontal)	TMT Angle (AP)	TMT Angle (Lateral)	TMT Angle (Axial)	TMT Angle (Sagittal)	TMT Angle (Frontal)	Sagittal Lift (mm)	Meary’s Angle
Conti (2020) [23]	HV	31	31	-	-	-	16.7 (10–25)	-	-	-	-	-	-	-	-
Lalevée (2022) [31]	Healthy	20	20	-	-	-	-	-	-	-	-	-	-	-	-
	HV	22	22	-	-	-	-	-	-	-	-	-	-	-	-
Kimura (2017) [36]	Healthy	10	10	-	-	-	9.3 (1.3)	-	-	-	-	3.2 (1.3)	-	-	-
	HV	10	10	-	-	-	22.1 (4.1)	-	-	-	-	6.5 (2.6)	-	-	-
Randich (2021) [40]	Healthy	36	36	-	-	-	-	8.77 (2.45)	-	-	-	-	−1.28 (6.33)	-	-
	HV	10	10	-	-	-	-	16.45 (4.47)	-	-	-	-	−5.36 (6.28)	-	-
Lee (2022) [14]	Healthy	30	30	-	-	9.46 (2.58)	-	-	-	-	-	0.23 (0.42)	-	-	-
	HV	27	30	-	-	16.98 (5.27)	-	-	-	-	-	1.15 (1.23)	-	-	-
Ji (2023) [27]	Healthy	-	79	-	-	8.3 (7.8–8.7)	-	-	-	-	-	0.9 (0.8–1.0)	-	-	-
	HV	-	82	-	-	14.8 (11.8–16.7)	-	-	-	-	-	1.6 (1.6–2.1)	-	-	-
Conti (2022) [22]	HV	39	-	15.6 (3.2)	-	-	-	-	-	-	-	-	-	-	-
				-	-	-	-	-	-	-	-	-	-	-	-
King (2004) [29]	Healthy	15	-	8 (2)	-	0.0001	-	-	8 (4)	4 (8)	-	-	-	0.3 (0.5)	-
	HV	25	-	15 (3)	-		-	-	11 (7)	13 (8)	-	-	-	2 (2)	-
Ferreyra (2022) [33]	HV	30	37	16.42	-	-	-	-	27.2 (7.3)	-	-	-	-	-	-
Kernozek (2002) [28]	HV	25	-	14.5 (1.7)	-	-	-	-	-	-	-	-	-	-	-
Naguib (2018) [38]	HV	-	59	23.86 (7.76)	-	-	-	-	-	-	-	-	-	-	-
Klemola (2017) [42]	HV	66	84	13.3 (2.7)	-	-	-	-	-	-	-	-	-	-	−3.7 (6.8)
Coughlin (2007) [24]	HV	103	122	14.5 (7–23)	-	-	-	-	-	-	-	-	-	-	-
Thompson (2023) [41]	HV	77	90	14.9 (3.1)	-	-	-	-	-	-	-	-	-	-	-
Ahuero (2019) [32]	HV	13	14	16 (9.5–21)	-	-	-	-	-	-	-	-	-	-	-
Faber (2001) [25]	HV	94	109	-	-	-	-	-	-	12.9 (4.8)	-	-	-	-	-
Oravakangas (2016) [39]	HV	20	23	17 (2)	-	-	-	-	-	-	-	-	-	-	−5 (8)
Greeff (2020) [26]	HV	23	32	15 (11–20)	-	-	-	-	-	-	-	-	-	-	-
Almaawi (2021) [43]	HV	89	100	14.4 (3.3)	-	-	-	-	-	-	-	-	-	-	5.5 (4.1)
Kopp (2005) [30]	HV	29	34	15.9 (10–22)	-	-	-	-	-	-	-	-	-	-	-
Ozturk (2020) [34]	HV	10	10	13.8 (0.5)	-	-	-	-	-	-	-	-	-	-	-
Manceron (2022) [35]	HV	-	20	13.3	-	-	-	-	-	-	-	-	-	-	-
	HV	-	20	14.8	-	-	-	-	-	-	-	-	-	-	-
	HV	-	9	16.9	-	-	-	-	-	-	-	-	-	-	-
Dayton (2020) [37]	HV	108	109	13.3 (2.4)	-	-	-	-	-	-	-	-	-	-	-

## Data Availability

The data presented in this study are available in Table 2 and Table 3.

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
