# Peer review of "Weightbearing Imaging Assessment of Midfoot Instability in Patients with Confirmed Hallux Valgus Deformity: A Systematic Review of the Literature"

_diagnostics, 2024, doi:10.3390/diagnostics14020193_

Round 1
Reviewer 1 Report
Comments and Suggestions for Authors
This study represents a systematic review that analyzes the instability of the mesofoot in patients with confirmed HVD, evaluated through weight-borne imaging modes, in order to deepen the understanding of how HVD affects the midfoot and potentially drive future treatment.
This systematic review has been conducted in accordance with the latest PRISMA (Preferred Reporting Items for Systematic Reviews) guidelines to ensure maximum clarity.
Data were extracted and articles classified using the Methodological Index for Non-randomised Trials (MINOR).
I read the article with interest, the title is well thought out and faithfully reflects the content of the study.
A) The abstract is sufficiently developed, and it is useful to frame the purpose of the study, but a few concerns are present:
Comment 1; It would be useful to better specify the results obtained
Comment 2; Better specify the purpose of the study
Comment 3; Clarification of the conclusions
B) In the introduction, the characteristics of the DDH have been sufficiently described.
Comment 1; deepen the explanation of the angles that are taken into account by the study
C) The materials and methods is well developed
Comment 1: Better specify inclusion criteria
Comment 2: Studies evaluating post - operator radiographic examinations have been considered?
D) The discussion is sufficiently developed.
E) Better explain the conclusion and goal of the study.
F) Nevertheless, some minor changes are needed to be considered suitable for publication.
Comments on the Quality of English LanguageMinor editing of English language are needed.
Author Response
Reviewer 1:
Thank you for carefully reviewing this manuscript and providing valuable feedback and insight. I have taken time to address your comments and concerns below.
Comment 1; It would be useful to better specify the results obtained.
I have added context into the results.
Lines 33-36:
“These findings validate the limited evidence thus far in the literature pertaining to consistent midfoot instability in HVD patients and is able to provide ample reasoning for physicians to place a larger emphasis on midfoot imaging when assessing HVD in its entirety.”
Comment 2; Better specify the purpose of the study
I added some additional context to this abstract.
Lines 19-20: “However, very few studies have placed a heavy emphasis on studying this phenomenon.”
Comment 3; Clarification of the conclusions
This comment has also been addressed above. Lines 33-36.
Comment 4; deepen the explanation of the angles that are taken into account by the study
I have added some information behind the reasoning for including these measurements.
Lines 46-48: “While intermetatarsal angle is often measured for patients with HVD, evidence is lacking to conclude that physicians factor its value into decisions.”
Lines 53-55: “For example, TMT angle taken from a sagittal view has shown significant correlations to HVD progression, likely due to causing increased dorsiflexion at the MTP joint.”
Line 55-56: “… such as hallux valgus angle”
Comment 5: Better specify inclusion criteria
I have added more detail to this section. I believe that everything besides the reasoning for 10 patients does not need much more detail.
Lines 84-85: “Ten patients was chosen as a patient threshold to prevent low power studies from po-tentially diluting the findings”
Comment 6: Studies evaluating post - operator radiographic examinations have been considered?
I included the reasoning behind this in the manuscript. Please see the added text. Post-operative values were not relevant for this study.
Lines 91-92: “As this study was solely focused on HVD midfoot instability, post-operative measurements were not relevant.”
Comment 7: Better explain the conclusion and goal of the study.
I have elaborated on this further in the conclusion.
Lines 110-113: “This review provides valuable evidence that verifies the little literature surrounding midfoot instability in conjunction with HVD. Future clinical care should place a large emphasis when not only diagnosis HVD, but also assessing the severity of deformity.”
Reviewer 2 Report
Comments and Suggestions for Authors
I commend the authors for their outstanding research effort as presented in the manuscript titled "Weightbearing Imaging Assessment of Midfoot Instability in Patients with Confirmed Hallux Valgus Deformity: A Systematic Review of the Literature." This systematic review aims to comprehensively summarize studies related to hallux valgus deformity (HVD), with a specific emphasis on understanding midfoot instability through weightbearing imaging assessments of the tarsometatarsal (TMT) joint. The subject matter explored in this study is captivating, and the manuscript achieves a commendable level of clarity and readability. The introduction is meticulously crafted, the results are presented clearly, the discussion is sound, and the conclusions are well-grounded in the findings.
However, a particular aspect of the manuscript requires further clarification. Notably, there is a substantial body of literature on the minimally invasive chevron Akin osteotomy (MICA) as a treatment for HVD. In the discussion section, it would be beneficial for the authors to incorporate a comment on MICA treatment to strengthen the manuscript. I encourage the authors to include relevant references, such as those available at doi:10.1186/s12891-023-06706-1, for more in-depth information.
In conclusion, while this manuscript shows promise, addressing the aforementioned point is crucial to elevate its impact on the field and enhance its suitability for publication.
Author Response
Reviewer 2:
Thank you for carefully reviewing this systematic review and providing valuable insight and comments.
Regarding your comment, I do not believe that conversation of any treatment modalities, including MICA, has much relevance to this review. While I agree that MICA has a growing body of informative studies, this review is simply focused with how midfoot measurements relate to HVD severity. Including information on how to treat HVD is beyond the scope of this review.